# A New Measure to Characterize the Degree of Self-Similarity of a Shape and Its Applicability

**DOI:** 10.3390/e22091061

**Published:** 2020-09-22

**Authors:** Sang-Hee Lee, Cheol-Min Park, UJin Choi

**Affiliations:** Division of Industrial Mathematics, National Institute for Mathematical Sciences, Daejeon 34047, Korea; mpcm@nims.re.kr (C.-M.P.); ujchoi@nims.re.kr (U.C.)

**Keywords:** branch length similarity (BLS) entropy, the degree of self-similarity, shape characterization, image analysis

## Abstract

We propose a new measure (*Γ*) to quantify the degree of self-similarity of a shape using branch length similarity (BLS) entropy which is defined on a simple network consisting of a single node and its branches. To investigate the properties of this measure, we computed the *Γ* values for 70 object groups (20 shapes in each group) in the MPEG-7 shape database and performed grouping on the values. With relatively high *Γ* values, identical groups had visually similar shapes. On the other hand, the identical groups with low *Γ* values had visually different shapes. However, the aspect of topological similarity of the shapes also warrants consideration. The shapes of statistically different groups exhibited significant visual difference from each other. Also, in order to show that the *Γ* can have a wide variety of applicability when properly used with other variables, we showed that the finger gestures in the (*Γ*, *Z*) space are successfully classified. Here, the *Z* means a correlation coefficient value between entropy profiles for gesture shapes. As shown in the applications, *Γ* has a strong advantage over conventional geometric measures in that it captures the geometrical and topological properties of a shape together. If we could define the BLS entropy for color, *Γ* could be used to characterize images expressed in RGB. We briefly discussed the problems to be solved before the applicability of *Γ* can be expanded to various fields.

## 1. Introduction

Image processing refers to the process of converting an actual physical image into a digital image and applying a variety of algorithms to distinguish between the target object we are interested in and other parts (background). Over the last few decades, the development of these algorithms has been cost-effective and rapid, and image processing has now become a technology used throughout the modern information society, consisting of multimedia systems. Most of these algorithms regard the target object as a shape (silhouette) [1,2] for the ease of extraction and analysis of the geometric properties of the object. This process of extraction and analysis is usually referred to as shape analysis.

This investigation can be generally divided into three types according to the approach used: skeleton-based, region-based and contour-based methods [3].

The skeleton-based method uses the area center axis of a shape to extract its features, which has the advantage of robustness in the case of occlusions and joints [4,5]. This method has been widely applied to characterize the dynamics of a target object in motion and to quickly determine a promising candidate in the image database in which the query is given. Region-based techniques, on the other hand, focus on a global analysis of images to extract their features [6,7]. This method has been used extensively; however, its performance is limited when the images to be distinguished are very similar in shape (e.g., leaves). Finally, the contour-based methodology considered in this study extracts the characteristics of a shape using only its contour information [8,9,10]. Some examples of this category include Fourier descriptors [11,12,13], curvature scale space [14], and multiscale fractal dimensions [15,16,17]. Most of the methods used in these cases are intuitive ways of dealing with the coordinate values of a series of points that form the boundary of a shape.

Lee et al. (2010) [18] first defined branch length similarity (BLS) entropy in networks with one node and multiple edges. Subsequently, Lee (2010) [19] showed that BLS entropy has a robust property when classifying shapes with noise at the shape boundary. These two studies showed with the battle tank shape that a small deformation in the shape causes a large change in the BLS entropy profile, and that it is invariant with respect to rotation, enlargement, and reduction of the shape. Based on the properties, Kang et al. [20] showed that when a network is constructed by connecting each pixel forming a shape boundary with all other pixels on the shape boundary, the set of BLS entropy values (the entropy profile) for the network created by each pixel is highly useful for characterizing the shape. The authors successfully classified the butterfly species by calculating the entropy profiles for the butterfly wing shape of various species and comparing the degree of correlation between the profiles. Lee et al. (2011) [21] constructed the facial networks on images of 70 male and 56 female faces displaying four different expressions (neutral, happy, angry, and screaming) by joining 17 facial landmark points such as the centers of the eyes, the corners of the mouth, and the underside tip. They calculated BLS entropy values for the networks and showed that the values were well grouped by emotion. Lee and Kang (2015) [22] used BLS entropy to quantify the behavior of *Canorhabditis* elegans in response to very low toxic substances. The authors set 10 equal coordinate points on the body of the worm and calculated the BLS entropy values for the angle and distance between the coordinate points. This study showed that the two entropy values could be very good indices for describing the movement of worms with elongated bodies such as *C*. elegans. Jeong et al. (2019) [23] recorded the movement of *C*. elegans and quantified the worm shape in each frame with the BLS entropy profile. The profiles were classified into 7 shape patterns through self-organizing maps combined with *k*-means clustering algorithm. By learning the classified patterns with two hidden Markov models, the authors developed a bio-monitoring system that determines water quality based on a series of shape patterns during a specific observation time.

Studies on the mathematical properties of this entropy were also conducted. Jeon and Lee (2012) [24] investigated the BLS entropy profile of a shape with infinite resolution and numerically investigated the variation in the pattern of the entropy profile caused by changes in the resolution in the case of finite resolution. Kwon and Lee (2014) [25] extended the theorem to the network created by linking infinitely many nodes distributed on the bounded or unbounded domain in **R**^k^ for *k* ≥ 1. However, the study of the mathematical properties of the entropy profile is still poor. Until now, most studies using BLS entropy have taken an approach to extracting features from the entropy profile. In fact, we do not know what the features reflects in the shape. On the other hand, there was no research on whether the shape characteristic information could be inferred from the entropy profile. This is the opposite approach to the approach taken so far by studies using BLS entropy. This inverse problem can be said to be one example of problems that need to be further understood mathematically for the BLS entropy. This inverse problem may seem like a problem that is directly linked to algorithms such as compression or transformation of 2D images. In this respect, it can be said that solving the inverse problem is to broaden the scope of applicability of the BLS entropy profile.

The goal of this study is to present a new measure (*Γ*) based on the BLS entropy profile that characterizes the self-similarity of a two-dimensional shape, and to understand the property of the *Γ* through its applications. Achieving this goal means that we are taking one step further towards solving the inverse problem. To understand a property of the *Γ*, we calculated the *Γ* values for 70 groups of objects in the MPEG-7 shape database (each group contains 20 shapes), performed statistical tests, and interpreted the results in terms of the self-similarity. In addition, we applied *Γ* to the finger gesture classification problem and showed that when *Γ* is used with other variables, its applicability can be wider. In the discussion section, we briefly discuss the problems that need to be solved before *Γ* can be applied in various fields that require image analysis technology, such as the industrial and medical sectors.

## 2. Materials and Methods

### 2.1. BLS Entropy and Its Profile

BLS entropy [18,19] was defined as the probability of branch length in a simple network consisting of a single node and several branches (Figure 1). This probability was defined as $p_{j}=L_{j}/\sum_{k=1}^{n}L_{k}$, where *n* is the number of branches in the network and *L_k_* represents the length of the *k*th branch (*k* = 1, 2, 3, …, *n*). 

Thus, the BLS entropy can be mathematically written as
(1)$$S=-\overset{n}{\underset{j=1}{\sum }}p_{j}\log(p_{j})/\log\left(n\right),$$

This entropy (*S*) has the property that the closer the branch lengths of the network are, the closer its value is to 1.0, and the lower the similarity, the closer the value is to 0.0 [18]. To demonstrate this property, we have provided an example where two branch lengths increased by the same length, and another example where only one of the three branches increased. In the former case, the *S* value increased, while in the latter case, the *S* value decreased. 

Figure 2 illustrates the formation of a BLS entropy profile using a square shape. Considering that the border of the square is composed of pixels, we can select one of the pixels, form a network that connects the selected pixel to all the other pixels on the border, and calculate the entropy value for the network (Figure 2A). Similarly, the entropy value of neighboring pixels connected to the selected pixel can also be easily calculated. Finally, we can obtain the entropy values for all connected pixels in a counterclockwise order, which is the entropy profile (Figure 2B). In the figure, the entropy values were normalized to fall in the range of [0, 1].

### 2.2. The Degree of Self-Similarity for a Shape

Figure 3 schematically shows the procedure for calculating the self-similarity for a shape using a pentagonal shape as an example. First, as mentioned in Figure 2, we obtain the entropy profile for the shape. Subsequently, we position a series of rectangular windows (see the red squares in Figure 3A) of appropriately small size (=20) on the entropy profile. In the preliminary study, we tested cases where the window size was less than 15 or greater than 30 when *lag* = 5, 10, 15, and 20. If the number of entropy values were too small (window size < 15), overfitting occurred, and if there were too many (window size > 30) values, the linearity for the distribution of entropy values was broken. All four *lag* values gave the same result. Considering this fact, we set the window size to 20 and the lag value to 20. These windows are cyclically shifted in the right direction by a certain length (*lag* = *φ*) (Figure 3A). Subsequently, we compute the linear regression coefficients (e.g., slope *w*) for the different entropy values of each window. Although the total entropy profile is strongly curved, the 20 entropy values belonging to each window appear to be almost linear. The linear slope for the distribution can be obtained using gradient descent method. In the consecutive windows obtained when *φ* = 20, the number of windows similar to the first window can be determined by comparing the *w* value of the first window with those of all other windows. Here, we define that two windows are similar to each other when the values *w*_1_ and *w*_2_ of the two windows meet the following condition: (2)when w1×w2>0,c={w1w2≥0.8 for w1<w2w2w1≥0.8 for w1>w2

Here, we denote the number of similar windows as $M_{0}^{1}$. Likewise, by comparing the *w* value of the second window and those of the other windows, the number of windows similar to the second window is determined and written as $M_{0}^{2}$. Thus, we can obtain $M_{0}^{1}$, $M_{0}^{2}$, …, $M_{0}^{K}$ (Figure 3B) where *K* is the number of windows. In the preliminary study, when we took *c* ≥ 0.95, the number of windows with the same slope was too small to calculate the self-similarity. On the other hand, when the *c* value was set too low (≤0.65), the number was too high, and it was difficult to trust the value. For this reason, we chose a *c* value of 0.8. Similarly, for *φ* values, we tested for 5, 10, 15, and 20, and all four cases showed statistically the same results. The *φ* value may need to be changed depending on the complexity of the shape, but it was sufficient to take *lag* = 20 for the shapes used in this study. After obtaining the set of {$M_{0}^{1}$, $M_{0}^{2}$,…, $M_{0}^{K}$}, we also consider the cyclic shift of window by *φ*. Similarly, we define the number of windows similar to the *i*-th window as $M_{\varphi }^{i}$ for *i = 1, …, K*. The set of {$M_{0}^{1}$, $M_{0}^{2}$,…, $M_{0}^{K}$} can then be extended to the following matrix:(3)$$\left(\begin{array}{c} M_{0}^{1}\;M_{0}^{2}\;M_{0}^{3}\;\ldots \;M_{0}^{K}\\ M_{\varphi }^{1}\;M_{\varphi }^{2}\;M_{\varphi }^{3}\;\ldots \;M_{\varphi }^{K}\\ M_{2\varphi }^{1}\;M_{2\varphi }^{2}\;M_{2\varphi }^{3}\;\ldots \;M_{2\varphi }^{K}\\ \vdots \;\vdots \;\vdots \\ M_{\theta }^{1}\;M_{\theta }^{2}\;M_{\theta }^{3}\;\ldots \;M_{\theta }^{K} \end{array}\right)$$
where *θ* = [*LCM* (window size, *φ*)/*φ*] −1. When calculating the number of similar windows, we exclude the case of comparison to itself, i.e., in the *j*-th row, the number of similar windows (*h*_j_) can be written as follows: (4)$$h_{j}=\overset{K}{\underset{k=1}{\sum }}\left(M_{j\varphi }^{k}-1\right)/2$$

Therefore, the ratio of the number of similar windows in the K(K−1)/2 pairs of windows ($\gamma_{j}$) can be defined as
(5)$$\gamma_{j}=\frac{2h_{j}}{K\left(K-1\right)}\times 100\;\left(\%\right)$$

Here, *k*= 1, 2, …, *K* and *j* = 0, 1, 2, … *θ*. The highest *γ* value best reflects the self-similarity of the entropy profile obtained for the pentagonal shape. Thus, the self-similarity can be finally written as follows:(6)$$\Gamma =\max\left(\gamma_{j}\right),\;j=1,2,\ldots ,K.$$

## 3. Results

### 3.1. Effect of Node Density of a Shape Boundary on the BLS Entropy Profile

When we draw an equilateral triangle on a computer, the number of nodes (pixels) on each side may be slightly different. The border of a horizontal or vertical side has a higher pixel density than oblique lines. This is due to the limitations of shape resolution. Therefore, it is necessary to understand the variation of *Γ* value with the node density. To this end, we constructed a single straight line by connecting two straight lines of the same length with different node densities to intuitively understand the density effect (Figure 4A). The number of nodes of the front straight line was *N*_1_, and the number of nodes of the rear straight line was *N*_2_.

When *N*_1_ = *N*_2_, the entropy profile was exactly symmetric. However, in the case of *N*_1_:*N*_2_ = 120:100, the entropy profile was relatively high in the region of low node density. For the case of *N*_1_:*N*_2_ = 140:100, the entropy profile tended to rise further in regions with lower node density. This can be illustrated through the example shown in Figure 1. When a node on a straight line with a low node density is connected to nodes on a straight line with a high density, the dispersion of its length distribution is reduced compared to the opposite case. To see if this density effect is also apparent in shape, we draw a triangle with three straight lines of the same length and different node densities (Figure 4B). The red circle represents the starting point of the entropy profile. The line formed by the two points *a* and *b* had 100 nodes, and the line consisting of points *b* and *c* and *a* and *c* had 120 nodes, respectively. The entropy profile was clearly higher in straight lines *a–b* with lower node density.

### 3.2. Application I: Statistical Analysis of Self-Similarity (Γ) in the MPEG-7 Image Database

To assess the visual similarity of two shapes with similar *Γ* values from a geometric point of view, we computed the *Γ* values for 1400 shapes in the shape database (MPEG-7) [26]. The computer hardware used for the computation had an Eight-core processor (3.6GHz) and RAM (32GB), and the software was MATLAB R2019a. The time taken to obtain the final result was approximately 4 h. The MPEG-7 database consists of 70 different object groups (20 shapes in each group). Subsequently, we performed one-way ANOVA and post hoc (Duncan) tests (Table 1). In Table 1, *N*, *M*, and *G* denote object groups, the average value of *Γ*, and statistical similarity indication, respectively. Considering the values of *M* and *G*, there were groups that belong to the same set with high *M* values, while there were also sets that are bound with low *M* values and sets that are loosely bound (e.g., *N* = 24, 47, 52, 65). The loosely bound groups appeared at levels where the *M* value was not too high or too low. This indirectly means that the shape classification from the *Γ* point of view is somewhat different from the existing method of simply classifying from the geometric point of view. Groups 19, 56 and 70 were statistically identical to each other with relatively high *Γ* values. The shapes belonging to these groups had the characteristic of being long in one direction and were visually similar (Figure 5). Conversely, groups 14, 27, and 28 had relatively low *Γ* values and were statistically equal to each other. The shapes belonging to groups 27 and 28 were visually similar to each other, and the shapes in group 14 were significantly different visually from those in the two groups. 

This was caused by the low *Γ* values. However, these shapes had a common topological structure in which several arms were attached to a circular body. This means that lower *Γ* values reflect topological attributes than the geometric attributes of the shapes. For the groups 24, 45, and 60 with low and statistically different *Γ* values, the shapes belonging to each group were completely different visually.

### 3.3. Application II: Self-Similarity (Γ) for the Classification of Finger Gestures

In this study, we applied the self-similarity (*Γ*) to the finger classification problem and showed that the *Γ* works well for this problem. It took about 5 min to solve this problem using MATLAB 2019a on the same computer used to analyze the MPEG-7 shape database. For this purpose, we obtained finger gesture shapes representing the numbers 1 to 10 from five different adults [27]. The number of shapes corresponding to each number was 10. Considering the *Γ* value for MPEG-7 image shapes, we reasonably inferred that for hand gesture shapes with a low *Γ* value, it would be difficult to distinguish the shapes from each other with only one *Γ* value. We, therefore, introduced *Z*, a new quantity based on the BLS entropy profile to improve discrimination between finger gesture shapes.

Before defining *Z*, when the entropy profiles for the two-finger gesture shapes are $P_{j}^{i}$ and $P_{n}^{m}$, we must first calculate the correlation coefficient, *Q*, between the two profiles. *Q* is mathematically described as follows:(7)$$Q_{j}^{i}\left(P_{n}^{m}\right)=\max\left\{corrcoeff\left(shifted(P_{j}^{i}),P_{n}^{m}\right)\right\}$$

Here, *i* refers to a group of finger shapes representing the number *i*, and *j* refers to the *j*-th shape in the group. *m* and *n* have the same meaning (*i*, *j*, *m*, *n* = 1, 2, 3, …, 10). The length of the entropy profile for each shape is 1,000; *max* {} represents the maximum value among the correlation coefficient values obtained for $P_{j}^{i}$ shifts by 1 (up to 1,000) and $P_{n}^{m}$. Considering that $Q_{j}^{i}$ corresponds to the branch length of the simple network shown in Figure 1, we define *Z* as the BLS entropy value for the network as follows:(8)$$Z^{i}=\frac{1}{10}BLS\{mean_{j}(Q_{j}^{i})\}$$

Here, the notation, “*BLS*” represents the entropy value for the mean values of $Q_{j}^{i}$ for *j* = 1, 2, …, 10.

Figure 6 shows a typical finger gesture shape that represents a number from 1 to 10 and shows the mean and standard deviations of the *Γ* and *Z* values of the shapes that belong to each number group. Gesture shapes that symbolize the numbers 1 to 5 are made by showing the back of the hand, while gestures that represent the numbers from 6 to 10 are implemented by showing the palm. Using loci coordinates of the little finger (pinky) and the longest finger (middle finger), we can easily distinguish between 1–5 and 6–10. In the lower figures, the center coordinate values of the square are the mean values of *Γ* and *Z*, respectively, while the width and height of the square represent the standard deviation values of *Z* and *Γ*, respectively. The lower left figure shows the results for a finger shape representing the numbers 1 to 5 (see the number next to each square), while the lower right figure shows the results for a finger shape representing the numbers from 6 to 10. In each figure, the squares were well separated from each other. This means that *Z* and *Γ* can be useful for resolving finger gesture recognition challenges.

## 4. Discussion

In this study, we proposed a new measure, the self-similarity of a shape (*Γ*), which characterizes shapes based on the BLS entropy profile. This characterization allows direct comparison of the degree of differences between shapes, e.g., judging which two shapes among three given shapes (see Figure 7) are more similar. Some people visually judge with confidence, while others judge differently depending on their perspective. However, by calculating the *Γ* value for the shapes (window size = 20, *lag* = 20, *c* = 0.8), we can clearly affirm that the shapes in the middle and right are more similar to each other, and even estimate that these two shapes are roughly 3% different from the shapes in the left. This shape comparison approach could be used to solve shape problems such as image retrieval. 

In Section 3.1, we suggested two issues that need to be addressed to better understand the self-similarity. One is the relationship between the number of nodes forming the boundary of the shape and the BLS entropy profile, and the other is the relationship between the vertices of the shape and the cusps of the entropy profile. In the former case, from the fact that the difference in the number of nodes of the shape boundary causes a very large change in its entropy profile, we easily infer that this problem is linked to the sensitivity of the entropy profile to shape deformation. Thus, further analysis of the node density effect would be needed as a follow-up study. In the latter case, as shown in Figure 4, the BLS entropy profile near the vertex position of an equilateral triangle showed cusps. We confirmed that these cusps appeared at various vertexes of different shapes. The sharpness and height of the cusps seemed to be strongly affected by the geometrical factors around the vertex. By understanding the cusp problem, we would take one step closer to solving the inverse problem of inferring the original shape from the entropy profile. 

For Section 3.2, researchers have conducted a study on the MPEG-7 shape database to classify 70 objects with various approaches. Largely, there are classifications using contour-based and region-based shape descriptors [26]. Also, contour-based descriptors can be divided into structural descriptors (e.g., chain code, polygons, Gaussian smoothing invariants) and convention descriptors (e.g., Fourier descriptors, perimeters, compactness). In the study of Zhang and Lu (2003) [28], the contour-based descriptor showed an accuracy of over 86.4% on average, while the region-shape descriptor showed an accuracy of over 76.5%. Some showed an accuracy of 95.0% or more depending on the type of descriptor. In recent years, fusion methods using artificial neural networks have been used to further improve the performance [27]. The existing methods mentioned above aim to improve classification accuracy. On the other hand, our method is not aimed at high classification accuracy, but rather shows which shapes are grouped according to the level of the *Γ* value. As shown in Figure 5, shapes belonging to statistically similar groups with high *Γ* values had visually similar characteristics, while shapes belonging to statistically identical groups with a low *Γ* value were geometrically similar or different. However, they were all somewhat similar topologically. In this respect, the existing method and our method cannot be directly compared in terms of the classification performance accuracy. Grouping shapes with a certain level of *Γ* value would allow us to select shapes that appropriately contain both geometric and topology information. From this point of view, it would be mathematically interesting to explore the relationship between the information and *Γ* values. 

For Section 3.3, Lee and Tanaka (2013) [29] proposed a simple algorithm for finger gesture recognition. The algorithm starts working with all fingers extended. Then, the smallest finger is set as the thumb. Then, the finger order is determined using the distance information between the fingers. Finally, the gesture is classified by checking whether each finger is folded or not. The authors tested the algorithm in dark, rough, and normal conditions, and obtained 97.1%, 96.7%, and 98.3% accuracy, respectively. There was also active research using neural networks [30] and support vector machines [31]. The studies had the advantage that the data preprocessing process was simple. For this problem, our method, a combination of *Γ* and *Z* based on BLS entropy, yielded approximately 92–95% classification accuracy. We photographed the hands of five people several times to get the shape of the hand. Among the obtained images, there were cases where finger gestures were made well, and cases were not. The results obtained from a well-made gesture shape yielded higher accuracy, and a relatively poorer gesture shape resulted in lower accuracy. Also, the intrinsic property of *Γ* contributed to the degradation of our method. The *Γ* cannot distinguish between symmetrical shapes. In this application example, *Γ* could not distinguish between the right hand and the left hand, and also could not clearly distinguish between the gesture shape when one index finger was incompletely straightened and the gesture shape when one little finger was completely straightened. To improve our method, it is necessary to introduce an additional variable that breaks the symmetry. In terms of shape classification accuracy, our method is inferior to the existing method. However, this comparison does not devalue the results in Section 3.3 in that a simple combination of Γ and Z showed not bad classification accuracy (more than 90%). In other words, our method showed room for performance improvement by adding other variables besides *Γ* and *Z*. 

One more issue that needs to be considered to be a follow-up study of this paper is to measure the self-similarity of shapes with colors or gray levels. One idea is to represent a shape with RGB channels as two BLS entropy profiles. One is an entropy profile that defines the distance between nodes as an edge, and the other is an entropy profile that defines a color value of nodes as an edge. Since two BLS entropy profiles can be created for each channel, a shape with RGB colors can be characterized by six BLS entropy profiles. The six *Γ* values calculated from the profiles could characterize the color shape. It would be very interesting to explore this problem. 

Consequently, we believe that this study is not only meaningful in defining and introducing a new concept of self-similarity based on BLS entropy, but also showing its applicability. In addition, we consider it worthwhile to propose problems that need to be addressed to broaden the applicability of self-similarity.

## Figures and Tables

**Figure 1 entropy-22-01061-f001:**
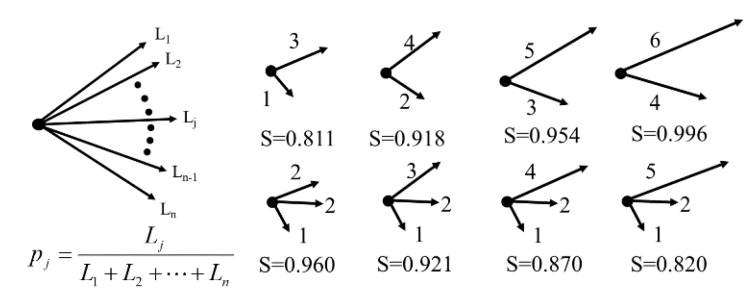
Length probability definitions for a simple network with one node and edges and examples.

**Figure 2 entropy-22-01061-f002:**
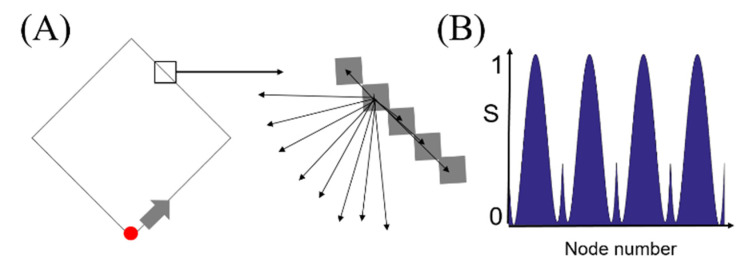
(**A**) Section of the shape outline magnified to display the individual pixels joined to every other pixel along the outline of a rectangle shape in order to form a node and branches, and (**B**) the BLS entropy profile for the shape.

**Figure 3 entropy-22-01061-f003:**
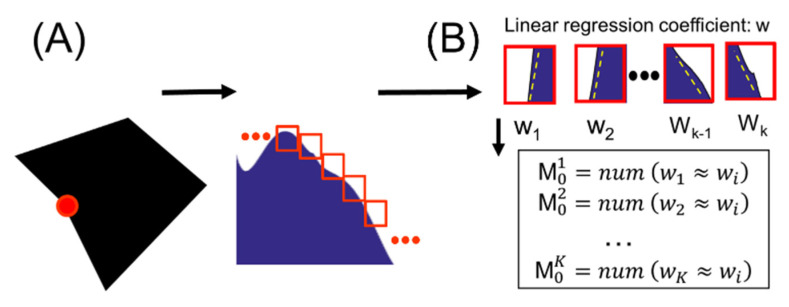
(**A**) The BLS entropy profile obtained from nodes forming a boundary of a shape (**B**) Brief process of calculating the degree of self-similarity of a shape from the entropy profile.

**Figure 4 entropy-22-01061-f004:**
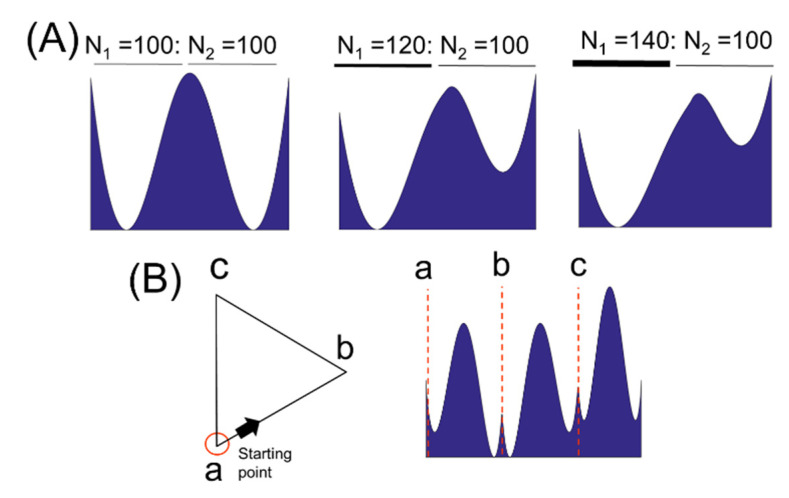
(**A**) BLS entropy profile for a straight line created by connecting two straight lines of the same length and different pixel density. Here, *N*_1_ and *N*_2_ represent the number of pixels of the two straight lines, respectively. (**B**) BLS entropy profile created by an equilateral triangle consisting of one side with low pixel density and two sides with high pixel density.

**Figure 5 entropy-22-01061-f005:**
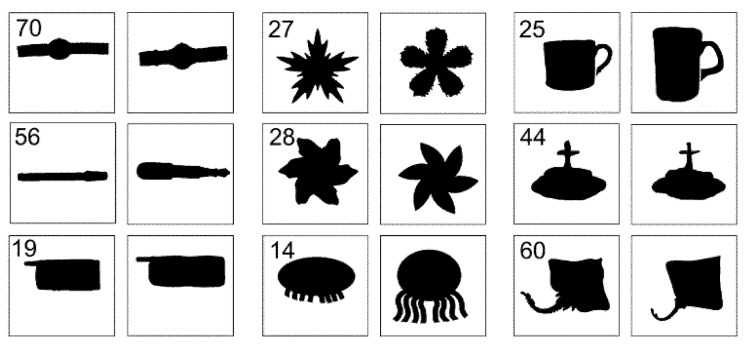
Two representative shapes belonging to several of the 70 object groups in the MPEG-7 shape database. Groups 70, 56, and 19 are the same, and groups 27, 28, and 14 are the same. Groups 26, 44, and 60 are different.

**Figure 6 entropy-22-01061-f006:**
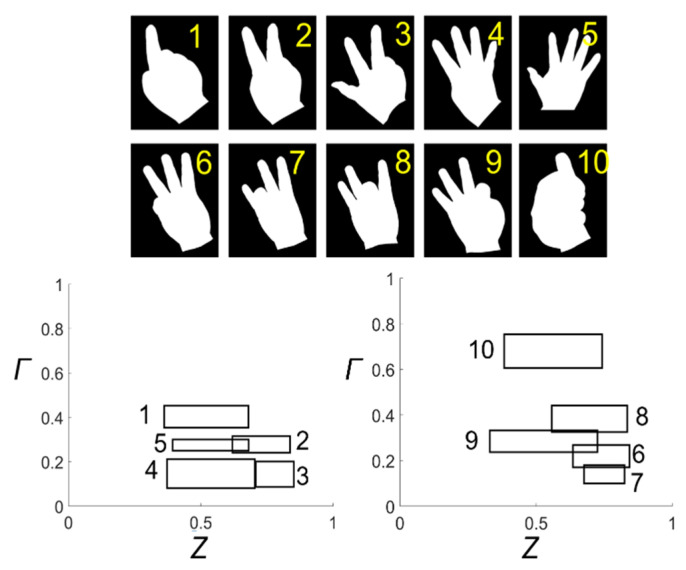
Finger gesture shapes symbolizing numbers 1 to 10 (upper) and the mean and standard deviation of *Γ* and *Z* for each shape (lower). The center coordinate values and length (height) of the rectangle represent the mean and standard deviation of *Γ* and *Z*, respectively.

**Figure 7 entropy-22-01061-f007:**
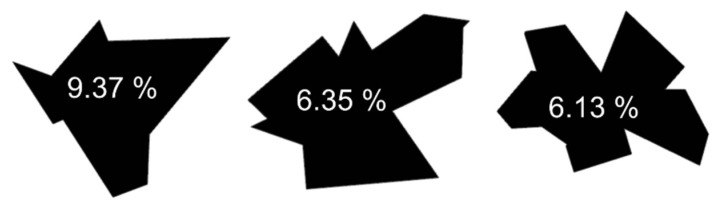
Three randomly generated polygons with 10, 15, and 17 vertices. The number in each shape represents the *Γ* value of the shape itself.

**Table 1 entropy-22-01061-t001:** One-way ANOVA and post hoc (Duncan) test results for the *Γ* values of 70 groups of objects (20 shapes in each group) included in the MPEG-70 shape database. Here, *N*, *M*, and *G* indicate the group number, the average *Γ* value of the group, and the presence or absence of significant differences between the groups, respectively.

N	M	G	N	M	G	N	M	G	N	M	G	N	M	G
70	0.1019	a	12	0.0895	fg	49	0.082	no	21	0.0761	tuv	45	0.0703	CD
56	0.1019	a	16	0.0891	fgh	67	0.0818	no	11	0.0757	tuvw	48	0.0703	CD
19	0.1009	a	47	0.0888	fghi	40	0.0815	o	53	0.0754	uvwx	43	0.0693	D
1	0.0987	b	24	0.0886	fghi	63	0.0807	op	58	0.0746	vwxy	14	0.067	E
39	0.0963	c	65	0.0879	fghij	26	0.0804	opq	6	0.0739	vwxyz	55	0.0669	E
62	0.0949	cd	52	0.0877	fghijk	4	0.0802	opq	38	0.0736	wxyzA	27	0.0665	E
30	0.0949	cd	46	0.0874	ghijk	8	0.0798	opqr	29	0.0732	xyzAB	28	0.0663	E
35	0.0938	de	32	0.087	ghijk	10	0.0797	opqr	20	0.0723	yzABC	50	0.0654	E
2	0.0936	de	31	0.0869	hijk	17	0.0788	pqrs	37	0.0721	zABC	66	0.0625	F
23	0.0935	de	22	0.0863	ijkl	64	0.0781	qrst	59	0.0719	zABC	25	0.0594	G
13	0.0929	de	69	0.0858	jklm	54	0.0777	rstu	51	0.0717	zABCD	44	0.0561	H
34	0.0926	de	57	0.0853	klm	18	0.0776	rstu	42	0.0714	zABCD	60	0.0536	I
36	0.0924	e	61	0.0843	lm	41	0.0773	stu	15	0.0712	ABCD	3	0.0522	I
68	0.09	f	9	0.0839	mn	5	0.0763	tuv	33	0.0707	BCD	7	0.0501	J

In the string of *G*, it means that groups containing the same alphabetic character are statistically the same, and are different between groups without the same character.

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
