# Peer review of "A New Measure to Characterize the Degree of Self-Similarity of a Shape and Its Applicability"

_entropy, 2020, doi:10.3390/e22091061_

Round 1

Reviewer 1 Report

In this paper is presented a “measure (Γ)” to characterize the degree of self-similarity of a shape based on the BLS entropy profile. The features have been computed for 70 object groups in the MPEG-7 shape database and performed statistical analysis on the results. The idea of this study is interesting, but unfortunately, this reviewer fails to see the main contributions. Specific comments are presented below:

  1. The current abstract only describes the general purposes of the article. It should also include the article's main (1) impact and (2) significance, mainly (Future Potential Techniques, for instance);
  2. Introduction. A more rigorous investigation on the existing methods (2013 to 2020), such as a comparison of previous approaches in terms of pros and cons, should be given. Since the proposed approach includes well-known methods more emphasis must be addressed on the novelty of the proposed technique. The authors referred mostly to ideas from consolidated techniques. For instance, a summary table can be used in this regard. Moreover, the main goals and contributions (in detail) should be presented in the last paragraph. The real contributions are difficult to recognize;
  3. That said, all parameters used to present the application should be indicated and discussed, as well as their tuning and effect on the results. On the other hand, parameters were defined/used without technical explanation, such as those indicated in subsection 2.2 (“…two windows are similar to each other when the values w1 and w2 of the two windows meet the following condition:..”);
  4. Experiments and results. The proposed method was tested on two datasets (Applications I and II). The authors should carry out further tests and present the results considering more categories of images, with cases involving different patterns (color and gray levels);
  5. From the results (subsections 3.2 and 3.3), a frank account should be provided with the strengths and weaknesses of the proposed research method. This should include a theoretical comparison to other approaches in the field, with the main advantages and limitations. Statistical tests can be used for validating the results, whose assumptions were not all met, such as more detail about the data, distributions and dependence, presence of significance outliers;
  6. Also, it is required to provide some including remarks to further discuss the proposed method, for instance: the main advantages and limitations in comparison with existing methods; are there other ways that the results can be further improved? The comparisons in the experiment are too simple. I hope the authors can refer to some more consolidated methods;
  7. The conclusion section seems to rush to the end. The authors need to rewrite the entire conclusion section with a focus on both impact and insights of the manuscript (for instance, the impact of the results for applied systems). No bullets should be used in your conclusion section. The authors will need to address your research contributions in theory.

Author Response

Response to Reviewer Comments

Reviewer 1

[1] The current abstract only describes the general purposes of the article. It should also include the article's main (1) impact and (2) significance, mainly (Future Potential Techniques, for instance);

[ANS] In the abstract, we mentioned the impact, significance, and future potential (see line 13-15, 19-24).

[2] Introduction. A more rigorous investigation on the existing methods (2013 to 2020), such as a comparison of previous approaches in terms of pros and cons, should be given. Since the proposed approach includes well-known methods more emphasis must be addressed on the novelty of the proposed technique. The authors referred mostly to ideas from consolidated techniques. For instance, a summary table can be used in this regard. Moreover, the main goals and contributions (in detail) should be presented in the last paragraph. The real contributions are difficult to recognize;

[ANS] Following the reviewer's point, we added details in the introduction section for studies after the concept of BLS entropy was proposed (lines 49-72). And we emphasized the pros and cons and novelty of the concept of self-similarity proposed in this study (lines 78-87). We also mentioned the main goals and contributions of this paper in the last paragraph (lines 88-97).

[3] That said, all parameters used to present the application should be indicated and discussed, as well as their tuning and effect on the results. On the other hand, parameters were defined/used without technical explanation, such as those indicated in subsection 2.2 (“…two windows are similar to each other when the values w1 and w2 of the two windows meet the following condition:..”);

[ANS] We added a description of the parameters used in section 2.2 and how the variable values were determined. (line 133-138, 153-157).

[4] Experiments and results. The proposed method was tested on two datasets (Applications I and II). The authors should carry out further tests and present the results considering more categories of images, with cases involving different patterns (color and gray levels);

[ANS] As the reviewer pointed out, it would be very interesting to add an analysis of the shapes with colors as well. However, we left this issue as a follow-up study to ensure that the focus of this paper is not entangled. We discussed in the discussion section the idea of overcoming the limitations of this study (applicable to silhouette shapes only) and dealing with colored shapes (line 356-362).

[5] From the results (subsections 3.2 and 3.3), a frank account should be provided with the strengths and weaknesses of the proposed research method. This should include a theoretical comparison to other approaches in the field, with the main advantages and limitations. Statistical tests can be used for validating the results, whose assumptions were not all met, such as more detail about the data, distributions and dependence, presence of significance outliers;

[ANS] We mentioned the advantages and disadvantages of our method in the application examples shown in Sections 3.2 and 3.3. And we discussed the limitations of our method caused by a lack of understanding of the relationship between the BLS entropy profile and shape (line 317-355).

[6] Also, it is required to provide some including remarks to further discuss the proposed method, for instance: the main advantages and limitations in comparison with existing methods; are there other ways that the results can be further improved? The comparisons in the experiment are too simple. I hope the authors can refer to some more consolidated methods;

[ANS] In this study, we defined self-similarity to shape and applied this concept to two application problems. The purpose of this paper is not to show that our method has better classification performance compared to the existing methods. What we want to emphasize is to show the nature of “self-similarity” and its applicability. In fact, our method is not an effective way to simply classify a group of 70 objects in the MPEG-7 database, but to classify it in terms of shape self-similarity. We added a description for this. From this point of view, we mentioned ideas that can further expand the applicability of Γ than ideas that further improve the classification accuracy of shapes. In addition, we also added an explanation throughout the discussion section about the limitations of our method (line 305-356)..

[7] The conclusion section seems to rush to the end. The authors need to rewrite the entire conclusion section with a focus on both impact and insights of the manuscript (for instance, the impact of the results for applied systems). No bullets should be used in your conclusion section. The authors will need to address your research contributions in theory.

[ANS] We revised the discussion section in accordance with the opinions of the reviewers. The strengths and weaknesses, limitations and potentials of our method were comprehensively mentioned.

Reviewer 2 Report

In this work, the degree of self-similarity of a shape is computed through a new measure proposed by the Authors. The proposal is investigated by using the MPEG-7 database. As an application, finger gestures are classified. Results demonstrate the potential use but some limitations are also discussed. Although the topic is interesting, some issues have to be addressed to clarify contribution and highlight applicability.

Please comment on the manuscript about the following.

In line 60, it is said that the mathematical properties of the entropy profile are still poor. Please be more specific and give examples of needs.

It is not clear the computation of linear regression coefficients.

In line 106, how do you determine the proper size of a window? Is it manually determined? What effect do different sizes have in the result? The same for “lag” (line 107).

Please number the equations. In line 115, how do you determine 0.8? If the value changes, what happens? Different tests have to be provided to justify the selection.

Section 3 Results includes the subsection 3.1 that shows the effect of node density of a shape boundary for the BLS entropy profile, but the BLS entropy profile has been already presented as stated in section 2.1. So, why is it analyzed in section 3.1? is there something new? Do you include some contributions on this topic?

Table 1 needs a major description.

In section 3.3, the classification of finger gestures is presented in order to show the applicability of your proposal but advantages over other methods are not discussed. Finger gesture recognition has been addressed by other methods, even your method has to be combined with other approaches (which have to be innovated) to be functional. Why is your method better? Why should your method be used?

Include the computational cost and the software used.

Change Fig. 7’s position.

To address the statement of line 254, the results for a triangle, polygons with more vertices, and a polygon almost circular should be included.

 It is not clear how the values of section 2.2. are used in your tests. In fact, why were such values used? What effect do different values have on the results of Table 1 and Figure 6. More experimentation is needed.

Author Response

Response to Reviewer Comments

Reviewer 2

[1] In line 60, it is said that the mathematical properties of the entropy profile are still poor. Please be more specific and give examples of needs.

[ANS] We described studies on the BLS entropy profile and mentioned the limitations of these studies (line 49-72). In addition, we give an example of a lack of understanding of the mathematical properties of the BLS entropy profile (line 78-87).

[2] It is not clear the computation of linear regression coefficients.

[ANS] Although the total entropy profile is strongly curved, the 20 entropy values belonging to each window appear to be approximately linear. The linear regression coefficient (the slope for the distribution of the 20 entropy values) can be obtained using gradient descent method (line 140-142).

[3] In line 106, how do you determine the proper size of a window? Is it manually determined? What effect do different sizes have in the result? The same for “lag” (line 107).

[ANS] In this study, gamma was calculated using an MPEG-7 shape database. In this case, the window size was set to 20 and lag = 0 (107 rows). To determine these values we tested cases where the window size was less than 15 or greater than 30 when lag = 0, 5 and 10. If the number of entropy values is too small (window size <15), overfitting occurs and too many values (window size> 30), the linearity is broken. All three lag values gave the same result. Considering this fact, we set the window size to 20 and the lag value to 0 (line 132-138).

[4] Please number the equations. In line 115, how do you determine 0.8? If the value changes, what happens? Different tests have to be provided to justify the selection.

[ANS] We numbered the equations in the paper. In this study, we assumed that the two slopes (w1 and w2) are equal if the c value is greater than 0.8. In the preliminary study, when we took c ≥ 0.95, the number of windows with the same slope was too small to calculate the self-similarity. On the other hand, when the c value was set too low (≤ 0.65), the number was too high, and it was difficult to trust the value. We confirmed that the self-similarity value is stable in the range of 0.75 <c <0.85. We chose the median value of c within this range (line 153-157).

[5] Section 3 Results includes the subsection 3.1 that shows the effect of node density of a shape boundary for the BLS entropy profile, but the BLS entropy profile has been already presented as stated in section 2.1. So, why is it analyzed in section 3.1? is there something new? Do you include some contributions on this topic?

[ANS] The square shape shown in Figure 2.1 has the same number of points (nodes) on each side. In this case, the entropy profile shows exactly periodic symmetry as shown in Fig. 2(b). On the other hand, as shown in Fig. 4(b), when the number of points forming a side is different, the symmetry of the entropy profile is broken. This is the node density effect. In Section 3.1, we did not do a deep mathematical analysis of the node density effect, but emphasized that the node effect can play an important role in understanding the relationship between the entropy profile and its shape.

[6] Table 1 needs a major description.

[ANS] We added the major description for Table 1 (lines 218-224).

[7] In section 3.3, the classification of finger gestures is presented in order to show the applicability of your proposal but advantages over other methods are not discussed. Finger gesture recognition has been addressed by other methods, even your method has to be combined with other approaches (which have to be innovated) to be functional. Why is your method better? Why should your method be used?

[ANS] In the discussion section, we added the discussion of Section 3.3, comparison with other research results, and explanation of the significance of our method (line 335-362).

[8] Include the computational cost and the software used.

[ANS] We performed MPEG-7 data analysis using MATLAB R2019b on a computer with an Eight-core processor (3.6GHz) and RAM (32GB). It took approximately 4 hours to obtain the results (see lines 214-216). On the other hand, the finger classification result took about 5 minutes (see line 250-252).

[9] Change Fig. 7’s position.

[ANS] We repositioned Fig. 7.

[10] To address the statement of line 254, the results for a triangle, polygons with more vertices, and a polygon almost circular should be included.

[ANS] We revised the discussion section as a whole, and the sentence pointed out by the reviewer was deleted.

 [11] It is not clear how the values of section 2.2. are used in your tests. In fact, why were such values used? What effect do different values have on the results of Table 1 and Figure 6. More experimentation is needed.

[ANS] We explained why the values of the variables (window size, lag, and c) mentioned in Section 2.2 were set. And it was mentioned that when the values of these variables change within an appropriate range, there is no effect on Table 1 and Fig. 6 (line 133-138, 153-157).

Reviewer 3 Report

This is an interesting study but needs an extensive effort to be considered for publication. There are many incomplete and confusing sentence. What do you mean by the first line of abstract: “in a previous study” which study are you talking about?!

Again, what do you mean by “Most subsequent studies”. What are you referring to? This is a wrong application of terms!

You mentioned statistical analysis, looking for the statistical method that you implemented but I cannot find anything? You need to explain what statistical method you used in a description of your method?

Also, the title needs to be changed, it is not as comprehensive as it should be and it is missing something

What do you mean by “Statistically identical groups”. Again, what is the method?

Lines 56-57, very awkward and wrong. You use investigate as a verb twice in a sentence

You finish your introduction without talking about a gap you are trying to bridge. Problem statement, objective and study contribution should be added in separate subsections so readers could get on board with what you are trying to achieve.

Why equations are not numbered? And some of them are in the middle of a line and some are at the sides?! They need to be fixed. For instance, equation between line 73 and 74 is messed up and explanation of the parameters is in the middle of the line

Why figures titles are so long. They should be concise and should be elaborated in the content of your manuscript, look at Figure 1, for instance

Line 105, (see the red square)! Where is the red square?! Which Figure?

In the discussion section you should talk about problem statement, what other studies done and what have you achieved. The future direction should be highlighted as well.

Author Response

Response to Reviewer Comments

Reviewer 3

[1] This is an interesting study but needs an extensive effort to be considered for publication. There are many incomplete and confusing sentence. What do you mean by the first line of abstract: “in a previous study” which study are you talking about? Again, what do you mean by “Most subsequent studies”. What are you referring to? This is a wrong application of terms!

[ANS] We corrected the sentence pointed out by the reviewer. Following the reviewer's comment, we modified the sentence to remove the ambiguity (line 9-11).

[2] You mentioned statistical analysis, looking for the statistical method that you implemented but I cannot find anything? You need to explain what statistical method you used in a description of your method?

[ANS] We added a description of the statistical method we used (line 218).

[3] Also, the title needs to be changed, it is not as comprehensive as it should be and it is missing something

[ANS] We modified the title slightly to suit the purpose of this paper.

[4] What do you mean by “Statistically identical groups”. Again, what is the method? Lines 56-57, very awkward and wrong. You use investigate as a verb twice in a sentence.

[ANS] We modified the sentences pointed out by reviewer (line 13-15).

[5] You finish your introduction without talking about a gap you are trying to bridge. Problem statement, objective and study contribution should be added in separate subsections so readers could get on board with what you are trying to achieve.

[ANS] We revised the entire introduction according to the reviewer's opinion.

[6] Why equations are not numbered? And some of them are in the middle of a line and some are at the sides?! They need to be fixed. For instance, equation between line 73 and 74 is messed up and explanation of the parameters is in the middle of the line.

[ANS] We numbered the equations and modified the line that the reviewer pointed out.

[7] Why figures titles are so long. They should be concise and should be elaborated in the content of your manuscript, look at Figure 1, for instance.

[ANS] We shortened the length of the title of Fig. 1 according to the reviewer's opinion. The titles of the other figures are rather long, but we left them as they are for better understanding.

[8] Line 105, (see the red square)! Where is the red square?! Which Figure? In the discussion section you should talk about problem statement, what other studies done and what have you achieved. The future direction should be highlighted as well.

[ANS] We fixed the sentence. And to reflect the reviewer’s opinion, the discussion section was revised as a whole (line 305-366).

Round 2

Reviewer 1 Report

The authors have done a good job of addressing the comments from the previous review.

Reviewer 2 Report

Most of my comments and concerns were addressed. Why was the sentence related to the question [10] deleted? Improve the quality of figures (use the same font,  increase size, etc.)

Reviewer 3 Report

The authors addres smy concerns